# A Hybrid Model for Predicting Bone Healing around Dental Implants

**DOI:** 10.3390/ma13122858

**Published:** 2020-06-25

**Authors:** Pei-Ching Kung, Shih-Shun Chien, Nien-Ti Tsou

**Affiliations:** Department of Materials Science and Engineering, National Chiao Tung University, Ta Hsueh Road, Hsinchu 300, Taiwan; gong1014.mse06g@nctu.edu.tw (P.-C.K.); play_58032.05g@g2.nctu.edu.tw (S.-S.C.)

**Keywords:** dental implant, tissue differentiation, bone remodeling, mechano-regulation theory, short-term healing, long-term healing

## Abstract

Background: The effect of the short-term bone healing process is typically neglected in numerical models of bone remodeling for dental implants. In this study, a hybrid two-step algorithm was proposed to enable a more accurate prediction for the performance of dental implants. Methods: A mechano-regulation algorithm was firstly used to simulate the tissue differentiation around a dental implant during the short-term bone healing. Then, the result was used as the initial state of the bone remodeling model to simulate the long-term healing of the bones. The algorithm was implemented by a 3D finite element model. Results: The current hybrid model reproduced several features which were discovered in the experiments, such as stress shielding effect, high strength bone connective tissue bands, and marginal bone loss. A reasonable location of bone resorptions and the stability of the dental implant is predicted, compared with those predicted by the conventional bone remodeling model. Conclusions: The hybrid model developed here predicted bone healing processes around dental implants more accurately. It can be used to study bone healing before implantation surgery and assist in the customization of dental implants.

## 1. Introduction

Implant stability is one of the important indexes to determine dental implant survival rates in the clinic [1]. It is dominated by the bone healing around the surgery site. Bone healing is a series of complex physiological processes that involved the regulation of several tissue phenotypes. At the beginning of bone healing, micro-vessels, and new connective tissue form on the surface of the wound, which is collectively referred to as granulation tissue [2,3,4,5]. After the formation of granulation tissue, further tissue differentiation initiates. Cells then transfer into fibrous connective tissues, cartilages, and new bones according to biophysical stimulus [6,7,8]. The final stage of bone healing is referred to as bone remodeling, which is a lifelong process, where the skeletal system maintains a dynamic equilibrium, related to the regulation of osteoclasts and osteoblasts [9,10,11]. When the balance is disrupted by external forces, a new equilibrium state can be achieved spontaneously. Thus, according to the healing process mentioned above, tissue differentiation and bone remodeling stages have a great impact on the short-term and long-term stability of implants, respectively [12,13].

To efficiently predict the short-term stability of implants in advance, we aim to simulate the tissue differentiation process by using the mechano-regulation algorithm. The origin of the method is based on Pauwels [14] who first specified that distortional stress and hydrostatic compression dominate tissue differentiation. Carter et al. [15] implemented the theory into a finite element model (FEM), revealing the evolution of connective tissues. Prendergast et al. [16] modified the methods by adopting octahedral shear strain and fluid flow as the solid and fluid stimuli. Lacroix further improved the model by using poroelastic finite elements, which can describe the biological mechanism in bones more accurately [5]. In recent years, many studies have revealed the effect of the mechanical environment and the geometric design of implants on the performance of tissue differentiation [17,18].

Bone adapts itself based on its mechanical environment and loading conditions, greatly affecting the morphology of bone and long-term stability of implants [11,13,19]. Many scientists have developed numerical methods to describe the behavior of bone remodeling [20,21,22,23,24]. Carter et al. [15,25,26] proposed that bone apparent density is dominated by strain energy density and studied the energy transfer in hip stems. The internal changes in bone morphology and the aging of connective tissues affected by the external loads were also predicted by FEM. Huiskes et al. [27] adopted a similar approach and simulated the femoral cortex around intramedullary prostheses to reveal the relationship between stress shielding effect and bone resorption. The bone remodeling algorithm was then extended to predict the variation of bone apparent density after implantation treatment [28,29,30,31]. The algorithm was verified by computed tomographic (CT) images, showing a high degree of similarity [32]. Most of the studies assumed a simple initial state of the models, where uniform material properties were assigned around the implants [29,31,33,34], i.e., the short-term bone healing has no effect on bone remodeling results. However, short-term healing is crucial since bone remodeling is an iterative process, where different initial conditions may lead to different bone density distribution around dental implants.

In order to test the null hypothesis of the bone healing process in the conventional model, we proposed a hybrid algorithm that regards the procedure of bone healing as two stages: (1) the short-term stage which was simulated by a tissue differentiation model and (2) the long-term stage which simulated by a bone remodeling model. At the beginning of the tissue differentiation model, it was assumed that the wound was filled with granulation tissue. The mechano-regulation algorithm was then applied to determine the tissue phenotypes for the following time steps. Once a stable tissue differentiation has been reached, the current tissue distribution with the material properties, such as Young’s modulus, apparent bone density, and Poisson’s ratio, in callus around the implant then served as the initial condition for the bone remodeling model. Then, the resulting long-term distribution of Young’s modulus and the remodeling stimulus will be discussed. and compared with the results which were similar to those done by Chou et al. [29], where the effect of short-term tissue differentiation was not considered.

The objective of this study is to develop a hybrid model that can predict the stability of dental implants and the strength of the surrounding bones with consideration of both the short-term and long-term bone healing process. The results of the current work can reveal the effect of bone with different material properties on bone healing, providing useful information for dental clinics. Furthermore, the proposed model can be used to rapidly examine the morphology design of dental implants (such as implant radius, length, thread geometry) and the placement protocol (such as insertion angle and depth) to improve the osseointegration between implants and bones.

## 2. Materials and Methods

Figure 1 shows the flowchart of the current hybrid algorithm, including short-term and long-term bone healing models. The distribution of strain, fluid velocity, and stem cell diffusion in the initial model (*t* = 0) was firstly calculated by FEM. Granulation tissues then differentiated into various tissue phenotypes based on the mechano-regulation algorithm. Next, the rule of mixture and smoothing procedure [35] was applied to determine the updated material properties and the detail will be discussed in Section 2.2. After the short-term healing process finished, the distribution of tissue phenotypes and the corresponding material properties around the implants was obtained and assigned to the bone remodeling model at the initial state for simulating the long-term healing process. Where bone remodeling algorithm adjusted the bone apparent density of each element iteratively until the equilibrium state of the remodeling stimulus under the given loading condition was achieved. The procedures will be discussed in more detail in Section 2.2 and Section 2.3.

### 2.1. Three-Dimensional FEM Model

The current hybrid algorithm was applied to study the bone healing of the mandibular second molar (back teeth in the upper jaw), where the bone geometry, density, and other material properties were adopted based on Chou et al. [29]. The geometry of the bone structure was obtained by extruding a planar CT image with a thickness of 80 mm, as shown in Figure 2a. It was referred to as the bone-tooth system, consisting of a layer of cortical bone overlying on cancellous bone, and a natural tooth. The physiological stimulus of the healthy state, i.e., bone-tooth system, was served as the objective function (i.e., attractor stimulus) for the calculation of bone remodeling in the bone-implant-prosthesis system, details of the calculation will be introduced in the next section. Next, the bone-implant-prosthesis system replaced the tooth in the bone-tooth system by a prosthesis and a short implant with a size of 5.0 × 5.1 mm; the remaining region, i.e., the extraction socket, was filled with callus, as shown in Figure 2b.

The models of the two systems were built by a commercial finite element package ANSYS 18.0 (ANSYS, Inc., Canonsburg, PA, USA). The implant, tooth, and prosthesis were meshed by the built-in element type, SOLID185; the remaining parts of the tissues were meshed by CPT215, which allows the calculation of poroelastic material properties, such as the fluid velocity and pressure in the pores of the bones. There were approximately 138,000 elements and 94,500 nodes for both systems. To maintain the balance between computation time and accuracy, finer meshes were applied around the interfaces between bone and the tooth/implant, as shown in Figure 2. The interfaces were set to allow sliding with a friction coefficient of 0.3. A symmetry boundary condition was applied to the mesial side of the model. All of the nodes in the distal side were constrained in all degrees of freedom. A displacement of 10.5 μm was applied at nodes on the top of the tooth/prosthesis. This value was equivalent to a biting force of 100 N [36,37,38]; the angle of the displacement was set according to it used in Chou et al. [29]. Note that loading, setting, and properties of all materials, including bones, prosthesis, tooth, implant, and bone graft, used in the current work were based on Chou et al. [29] for comparison, as shown in Table 1. It is worth mentioning that, in the tissue differentiation process, the material properties of elements in the callus region transformed with iterations, i.e., they evolved according to the corresponding tissue phenotypes during the iteration process. Details of the mechanism of the mechano-regulation algorithm will be explained in the next section.

### 2.2. Mechano-Regulation Algorithm

The mechano-regulation algorithm proposed by Lacroix and Prendergast [5,35] was adopted in the current work to predict the distribution of tissue phenotypes. The procedures of the algorithm, including the calculation in each iteration, updates of material properties, and post-processing to the results, were implemented by MATLAB 2019 (The MathWorks, Inc., Natick, MA, USA). At the beginning of the calculation, elements in the callus region were set with the material properties of granulation tissues. According to the theory, tissue differentiation (TD) is induced by the combination of octahedral shear strain (γ) and fluid flow (ν) caused by external loads. It is referred to as biophysical stimulus $S_{\mathrm{TD}}$ such that:(1)$$S_{\mathrm{TD}}=\frac{\gamma }{a}+\frac{\nu }{b}$$
where a = 0.0375 and b = 3 μm/s are empirical constants. Then, the tissue phenotype for the next iteration can be determined based on the value of $S_{\mathrm{TD}}$ as shown in Table 2. The material properties of the tissue phenotype were updated accordingly.

The concentration of mesenchymal stem cells determined the level of the transition from granulation tissue to the other tissue phenotypes. The migration [39] and proliferate [40] of mesenchymal stem cell can be simplified as a classical isotropic diffusion, such that:(2)$$\frac{dn}{dt}=D\nabla^{2}n$$
where t is time; D is the diffusion coefficient; n is the current concentration of mesenchymal stem cell. The migration of stem cells started from the boundary of the extraction socket, which is called cells origin and marked by yellow lines in Figure 2b. At the last iteration, the concentration of the stem cell reached the maximal value. In the current work, D is set as $8.85\times 10^{-14}{\;m}^{2}/s$. Then, the effective material properties of tissues for the next iteration, including Young’s modulus, Poisson’s ratio, and permeability, can be obtained by a linear combination between the granulation tissue ($X_{g}$) and the differentiated tissue ($X_{d}$) as follows:(3)$$X_{mix}=\frac{n_{max}-n}{n_{max}}X_{g}+\frac{n}{n_{max}}X_{d}$$
where $n^{max}$ is the maximum concentration; *n* is the current concentration of stem cells determined by Equation (2); $X_{d}$ is material properties of the differentiated tissue phenotype shown in Table 1.

To avoid the instability and dramatic change of material properties between iterations, Lacroix and Prendergast [35] suggested a smooth procedure to average the material properties from the previous nine iterations, which can be written as:(4)$$X_{i}=\frac{1}{N}\left(X_{mix}+X_{i-1}+X_{i-2}+\ldots +X_{i-\left(N-1\right)}\right)$$
where N = 10; i is the current iteration number; $X_{mix}$ is the effective material properties calculated from Equation (3). Note that when the iteration number is i < 9, smoothing operation is applied to the iteration i to the first [35]. The short-term healing time was set as 70 days, which is the average healing period for implantation surgeries [41,42,43]. The short-term healing process and the differentiated tissue phenotypes can then be predicted.

### 2.3. Bone Remodeling Algorithm

After the numerical calculation for the short-term healing, long-term healing, i.e., bone remodeling (BR), occurred to alter the internal structure of bones and reach a new equilibrium state based on the mechanical environment. Huskies et al. [27] proposed a bone remodeling theory assuming the driving force of self-adaptive activity is determined by remodeling stimulus ($S_{BR}$, unit: J/kg), such that:(5)$$S_{BR}\left(\vec{r},t\right)=\frac{u\left(\vec{r},t\right)}{\rho \vec{(r},t)}$$ where u is strain energy density (unit: J/m^3^); ρ is the apparent bone density (kg/m^3^), *t* is time; and $\vec{r}$ is the position vector [27]. When the value of remodeling stimulus $S_{BR}$ is greater than the given threshold, bone formation occurred and Young’s modulus and bone density increase accordingly. On the contrary, when the remodeling stimulus $S_{BR}$ is less than the threshold, bone resorption occurred, and Young’s modulus and bone density decrease. In addition, Carter [25] stated that bones maintain a state of homeostasis when the remodeling stimulus is in a certain range, which is referred to as a “lazy zone.” Thus, the bone remodeling process can be expressed by nonlinear functions of the remodeling stimulus [27]:(6)$$\frac{d\rho }{dt}=\{\begin{array}{lll} A_{f}\left[S_{BR}-\left(1+s\right)K(\vec{r}\right){]}^{2}\;, & S_{BR}\geq \left(1+s\right)K\left(\vec{r}\right) & \left(\mathrm{Formation}\right)\\ 0, & \left(1-s\right)K\left(\vec{r}\right)<S_{BR}\leq \left(1+s\right)K\left(\vec{r}\right) & \left(\mathrm{Lazy}\;\mathrm{zone}\right)\\ A_{r}\left[S_{BR}-\left(1-s\right)K(\vec{r}\right){]}^{3}, & S_{BR}\leq \left(1-s\right)K\left(\vec{r}\right) & \left(\mathrm{Resorption}\right) \end{array}$$
where $A_{f}$ and $A_{r}$ are formation and resorption coefficients; s is the threshold of the lazy zone, which is set as 0.75 [44]; $K\left(\vec{r}\right)$ is the attractor stimulus induced by the biting force in the bone-tooth system, which is determined by Equation (5). The value of $K\left(\vec{r}\right)$ in the region of callus was set to 5, which is the average of the overall remodeling stimulus in bone element. It is worth noting that the rate of bone resorption is greater than that of bone formation based on clinical observations, resulting in a greater exponential term of resorption in Equation (6). The apparent density of a bone element m at the j^th^ iteration can be derived by integrating Equation (6) with a forward Euler method [27,45,46,47], such that:(7)$$\rho_{m}^{j}=\{\begin{array}{lll} \rho_{m}^{j-1}+A_{f}∆t\left[S_{\mathrm{BR}}^{j-1}-\left(1+s\right)K(\vec{r}\right){]}^{2}\;, & S_{\mathrm{BR}}^{j-1}\geq \left(1+s\right)K\left(r\right) & \left(\mathrm{Formation}\right)\\ 0, & \left(1-s\right)K\left(r\right)<S_{\mathrm{BR}}^{j-1}\leq \left(1+s\right)K\left(r\right) & \left(\mathrm{Lazy}\;\mathrm{zone}\right)\\ \rho_{m}^{j-1}+A_{r}∆t\left[S_{\mathrm{BR}}^{j-1}-\left(1-s\right)K(\vec{r}\right){]}^{3}, & S_{\mathrm{BR}}^{j-1}\leq \left(1-s\right)K\left(r\right) & \left(\mathrm{Resorption}\right) \end{array}$$
where ∆t is the time increment [21,48]. Young’s modulus (E, unit: GPa) of bone elements was associated with the corresponding apparent density based on the finding of Carter and Hayes [26], such that:(8)$$E=C\rho^{3}$$
where C is constant. Note that the integration coefficients $A_{f}∆t$ and $A_{r}∆t$ in Equation (7) were set as 1 × 10^−11^ and the constant was set as 3.79, based on the setting used in Chou et al. [29]. The value of Young’s modulus indicated the strength of the internal structure of bone, which affected the calculation at the next iteration. The average remodeling stimulus ($S_{ave})$ in each iteration was recorded and served as a measure of convergence of the model, such that:(9)$$S_{ave}=\frac{1}{N_{total}}\overset{N_{total}}{\underset{k=1}{\sum }}S_{k}$$
where $N_{total}$ is the total number of bone elements; $S_{k}$ is remodeling stimulus of the local bone element *k*. It is worth noting that long-term bone healing, i.e., bone remodeling, is a lifelong process and the bone system evolves to reach an equilibrium state according to its current mechanical environment. Thus, the time step used here is, in fact, a computational increment and is not associated with a real-life time scale. The end of the calculation depends on the convergence of $S_{ave}$ as mentioned above.

## 3. Results

### 3.1. Short-Term Healing and Tissue Differentiation

Tissue differentiation and the evolution of bone ingrowth around the implant were evaluated by the mechano-regulation model. Figure 3 shows the percentage of tissue phenotypes in each day during the short-term healing process. In the early stage of tissue differentiation, granulation tissues still existed in the callus region. This is because tissue differentiation was initiated when the concentration of stem cells above certain levels. At this stage, the inner callus region has relatively low concentration as the stem cells diffused from the boundary of the callus region. Moreover, it was found that the soft tissue with a higher biophysical stimulus ($S_{\mathrm{TD}}>1$) such as cartilage and fibrous tissue decrease with time while the bone tissue increased continuously until the middle of the differentiation process (i.e., around the 35th day). After the 35th day, bones possessed a certain degree of strength, i.e., higher Young’s modulus. This resulted in the decreasing values of bone stimuli, and thus, maturate and immature bones gradually became the dominant tissue phenotype in the entire callus region. Then, most of the immature bones transformed into maturate bones as the healing process was closed to the 70th day.

Details of the tissue phenotype at the specific days are shown in Figure 4. On the 4th day, more than half of the callus region remained as granulation tissue, as shown in Figure 4. It is observed that cartilages and fibrous tissues occurred around the threads in the middle part and the bottom of the implant, respectively. It is because the applied load caused the stress concentration around the threads and the bottom, giving higher biophysical stimulus to the elements in that region. Note that, although there were mature and immature bones, the effective material properties, such as Young’s modulus, were still close to that of granulation tissue based on Equation (3), due to the low concentration of stem cell on the 4th day. Then, it can be observed that, on the 10th day, granulation tissues gradually transformed into mature and immature bones as the concentration of stem cells increased with time. It is worth noting that cartilages accumulated at tips of the threads since stress concentration was partially released with the increasing maturity of the surrounding bones. On the 30th day, granulation tissues were disappeared entirely. Cartilages were mainly located at the lingual side because of the oblique biting force. Then, maturate and immature bones gradually dominated the entire system around the implant on the 50th to 70th days. It is worth noting that there were very few elements of bone resorption on the 10th day and were not shown in the current cross-section in Figure 4. In this way, the short-term healing pattern around the implant was obtained.

### 3.2. Bone Remodeling

Now consider two bone remodeling models. The first, i.e., the current model, adopted the result at the end of short-term healing predicted by the mechano-regulation algorithm as the initial state; the second was the conventional model based on bone remodeling algorithm with the assumption that the extraction socket was filled with bone graft and regarded as the initial state where the material properties were set as uniform. The setting of the second model followed the work done by the literature [29]. Both models shared the same distribution of the objective attractor stimulus $K(\vec{r}$) based on the natural tooth, as shown in Figure 5a and Figure 6a. Then, the bone remodeling models altered the bone apparent density of all the bone elements in the following iterations to achieve that objective distribution.

In the current model, the distribution of Young’s modulus of the initial state is shown in Figure 5d. It can be observed that certain regions in the callus in Figure 5d were in dark and light blue colors giving low and high Young’s modulus, ranging from 0.001 to 6 GPa based on Table 1, due to the non-uniform stem cell concentration and the presence of different tissue phenotypes. It is worth noting that the average value of Young’s modulus in that region was around 2 GPa, having a good agreement with the value used in the conventional model where uniform Young’s modulus was assumed, as shown in Figure 6d. The corresponding bone apparent density of each element in the callus region can then be obtained by Equation (8). The resulting model with the updated bone apparent density was then subjected to the biting force, giving the distribution of the remodeling stimulus ($S_{BR}$) for the 1st iteration, as shown in Figure 5b. The values are shown in Figure 5a,b were substituted into Equation (7) to obtain the updated bone apparent density for the next iteration. The corresponding distribution of Young’s modulus in the 1st iteration was shown in Figure 5e. It can be observed that bone regions with high Young’s modulus (above 12.18 GPa, colored in red) fully covered around the implant; most of the bones attached to the surface of the implant were also with non-uniform Young’s modulus (ranging from 3.04 to 6.08 GPa). The calculation continued until the 100th iteration was achieved. It is worth noting that the average remodeling stimulus $\left(S_{ave}\right)$ of the current model quickly converged around the 10th iteration, showing great stability. The converged values $S_{ave}$ for both the current and conventional models were identical. The distribution of $S_{BR}$ of the 100th iteration is shown in Figure 5c. It can be observed that most of the regions had the value of $S_{BR}$ closed to those of the target, i.e., the objective attractor stimulus $K\left(\vec{r}\right)$. The final state of bone remodeling gave the distribution of Young’s modulus, shown in Figure 5f.

The average Young’s modulus of the entire system predicted by the current model was around 4.77 GPa. It is worth noting that bone resorptions (colored in gray) occurred around the threads toward the lingual side and around the neck of the implant. The total volume fraction of bone resorption was 0.042%. Similar to the 1st iteration, bone regions with high Young’s modulus were still fully covered around the implant. Two additional high strength bone tissue bands connected to cortical bones were formed in the bottom right (the lingual side) and top left (the buccal side) of the implant, which provided extra supports and enhanced the stability of the implant.

Next, Figure 6d shows the distribution of Young’s modulus of the initial state adopted in the conventional model. It can be observed that constant Young’s modulus of 2 GPa in the callus in Figure 6d was assumed without considering the result of differentiating tissue phenotypes during the short-term healing. Then, the distribution of the remodeling stimulus ($S_{BR}$) for the 1st iteration can be determined as shown in Figure 6b. In the 1st iteration, the distribution of $S_{BR}$ was similar to that generated by the current model, apart from there was no high stimulus bands around the implant. However, such a small difference resulted in a very different distribution of Young’s modulus shown in Figure 6e compared with that in Figure 5e, wherein, the high strength bones partially covered around the implant; most of the bones attached to the surface of the implant remained a constant Young’s modulus of 2 GPa. Then, similar to the current model, the average remodeling stimulus $\left(S_{ave}\right)$ quickly converged. Finally, the 100th iteration was achieved, giving the distribution of $S_{BR}$ as shown in Figure 6c. The resulting Young’s modulus distribution is shown in Figure 6f. The volume fraction of bone resorption was at the value of 0.044%, which was very similar to the current model. A significant difference between the results of the two models was the location of bone resorption. It was found that bone resorption (colored in grey) occurred in the buccal side in the conventional model while it appeared on the lingual side in the current model. This will be discussed in more detail in the next section. Another obvious difference between the two models was the distribution of high Young’s modulus bands. The high Young’s modulus band was absent in the buccal side, and thus, no supporting connection between the surface of the implant and cortical bone. The average Young’s modulus at the 100th iteration was around 3.65 GPa which was relatively lower than that generated by the current model.

## 4. Discussion

The results mentioned above show that the initial state of bone remodeling can greatly affect the distribution of Young’s modulus at the final state. In the current model, the initial state of bone remodeling was the result of the mechano-regulation model (short-term healing process), giving the top and bottom regions in the callus with lower Young’s modulus. This leads to a higher strain energy density and a low corresponding bone apparent density base on Equation (8). Then, these regions had higher remodeling stimulus based on Equation (5), promoting the formation of bones, i.e., Young’s modulus and apparent density increased. This non-uniform Young’s modulus at the initial state leads to a dramatic change of Young’s modulus in the entire callus region in the 1st iteration. On the contrary, in the conventional model, Young’s modulus was assumed uniform, resulting in the change of Young’s modulus around the implant only in the 1st iteration.

In the 100th iteration, it can be observed that the stability of the implants was greatly influenced by the initial states. In the current model, there were two high strength bone tissue bands connected to cortical bones, while there was only one connected bone tissue band in the result generated in the conventional model. In addition, the average Young’s modulus in the current model was higher than it predicted by the current model. Thus, the stability of the implant was underestimated in the conventional model where uniform Young’s modulus in the callus region at the initial state was assumed. This indicated that short-term healing can greatly affect the results of bone remodeling and cannot be neglected.

In the results of short-term healing generated by the mechano-regulation model, soft tissues occurred in the early stage and then replaced by bone tissues due to the decrease of biophysical stimulus in the later stage. This was in accordance with both the experimental [14] and computational [5] works in the literature. Where the literature reported that bone tissue forms after the formation of soft tissues (i.e., fibrous tissue and cartilage), and then bone began to differentiate, giving the increase of fluid flow since woven bone is more permeable. Furthermore, three additional features can be found in the remodeling result in the 100th iteration predicted by both the current and conventional models, which were in accordance with the clinical observations. Firstly, both models predicted a region of bone resorption at the top surface in the lingual (right) side. This phenomenon was known as marginal bone loss [49,50], which was an important factor for implant stability. Secondly, both models generated a bone resorption region in the middle of the bone-implant interface. This is the so-called stress shielding effect [27,47], which was the result of the occurrence of high strength bone tissue (colored in red) at the region around the first thread. However, the locations of the bone resorption region predicted by the two models were different, where the region predicted by the current model was in the lingual (right) side and that predicted by the conventional model was in the buccal (left) side. According to the literature [51], bone resorption occurred on the right-hand side when the loading direction was similar to that in the current work, i.e., the load was applied from the top right to the bottom left. The current model successfully captured this feature, reproduce the bone resorption region at the right-hand side at the bone-implant interface, while the conventional model predicted the location of bone resorption at the opposite side. This shows that the current model can give a more accurate result of bone remodeling procedure in the bone-implant-prosthesis system. Third, the result in the 100th iteration predicted by the current model shows that around 70% of the surface of the implant was covered by bone tissues (i.e., the elements with Young’s modulus greater than that of immature bone, 2 GPa). This value was similar to that reported by Lian et al. [31], where they suggested around 60% contact between bone and implant when an equilibrium of bone remodeling is reached. Based on the features mentioned above, we conclude the rejection of the null hypothesis that short-term bone healing has no effect on bone remodeling results.

Since the current hybrid model was implemented by the finite element method, which can perform virtual tests on a wide variety of people characteristics and dental materials by simply changing the geometry of the model, boundary conditions, and material properties of bones. Thus, the current model can potentially provide estimated information and may even provide optimized dental implants for dentist clinics. Next, to demonstrate the applicability of the current model and reveal the effect of individual differences of the patients, we adopted the case of middle-aged male adults with higher strength material property of bone, referred to as to higher bone strength case. Where Young’s modulus of cortical, cancellous, immature, and mature bone was 1.4 times [52] than it in Table 1, while the properties of the remaining tissues, such as fibrous tissue and cartilage, stayed unchanged. Then, the short- and long-term bone healing processes were evaluated by the current model. The corresponding results are shown in Figure 7 and Figure 8, respectively. It was found that the trend of tissue differentiation in the short-term healing process was almost identical to it the standard case as shown in Figure 4. The only difference between the two cases was the averaged Young’s modulus in the high strength case was about two times higher than it was in the standard case, which can be seen in Figure 8a, where the material properties in the callus region were assigned according to the tissue differentiation result. In the 100th iteration, there were several significant differences between the two cases. Firstly, there was merely no bone resorption in the higher bone strength case. The volume fraction of bone resorption was at a value of 0.0068%. Secondly, the connective tissue bands in the higher bone strength case were thicker than in the standard case. The two features indicate that the bone system in the higher bone strength case can provide excellent supports and enhance the stability of the implant. This result also has good agreement with the clinical observation, where the dental implant failure rate for the middle-aged male adults was relatively low [53].

Although the current model considered both the short-term and long-term healing process and reproduced many features that were discovered in the experiments, the model can be further improved by considering the physiological mechanism listed as follows. For example, a more complex diffusion mechanism of stem cell migration in the short-term healing process, such as the growth of vessels which can be implemented by the random-walk model [54]; periodontal ligaments (PDL), which play a crucial role in bone remodeling, can be simulated by taken the anisotropic and nonlinear elastic stress-strain behavior into account [55,56,57]. It is expected a more accurate prediction can be achieved if these factors were considered.

## 5. Conclusions

In this study, a hybrid numerical bone healing algorithm was developed to predict the morphology of bone around dental implants with the consideration of both short-term and long-term bone healing. The results showed that the effect of short-term bone healing should not be ignored, and the assumption of uniform material properties for the initial state in the bone remodeling model is inappropriate. The current hybrid model can reveal many bone healing features having a very good agreement with the literature. It can be extended to simulate different implant geometries, applied loads, and bone properties of patients, enabling an early prediction of the performance of clinical treatments.

## Figures and Tables

**Figure 1 materials-13-02858-f001:**
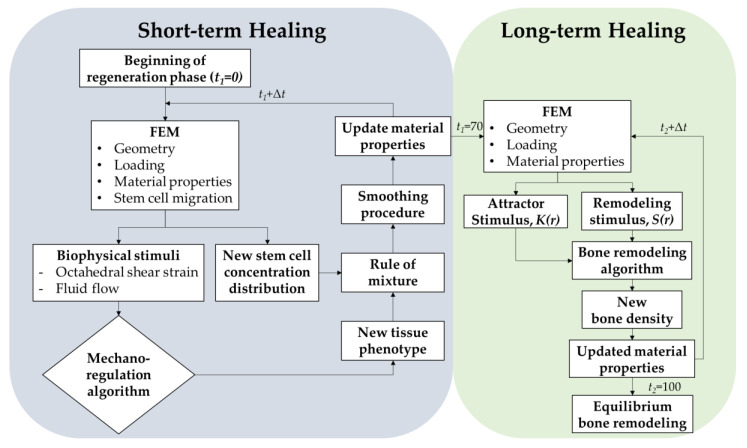
Flow chart of bone healing preoperative evaluation.

**Figure 2 materials-13-02858-f002:**
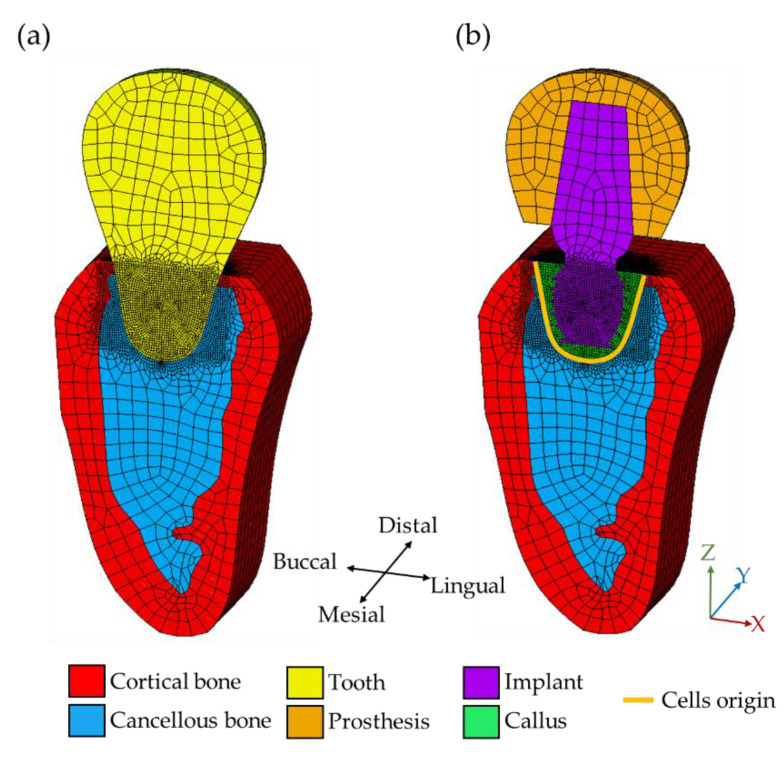
FEM model of (**a**) bone-tooth and (**b**) bone-implant-prosthesis systems.

**Figure 3 materials-13-02858-f003:**
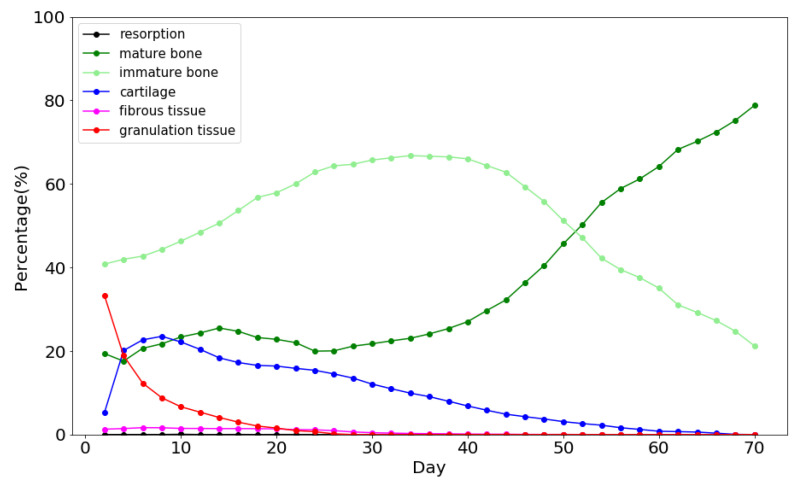
Percentage of various tissue phenotype in each day during the short-term bone healing process.

**Figure 4 materials-13-02858-f004:**
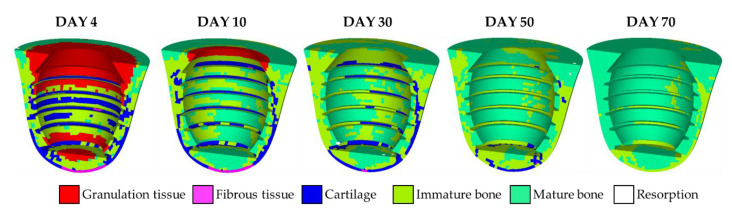
The tissue differentiation history predicted by the mechano-regulation algorithm.

**Figure 5 materials-13-02858-f005:**
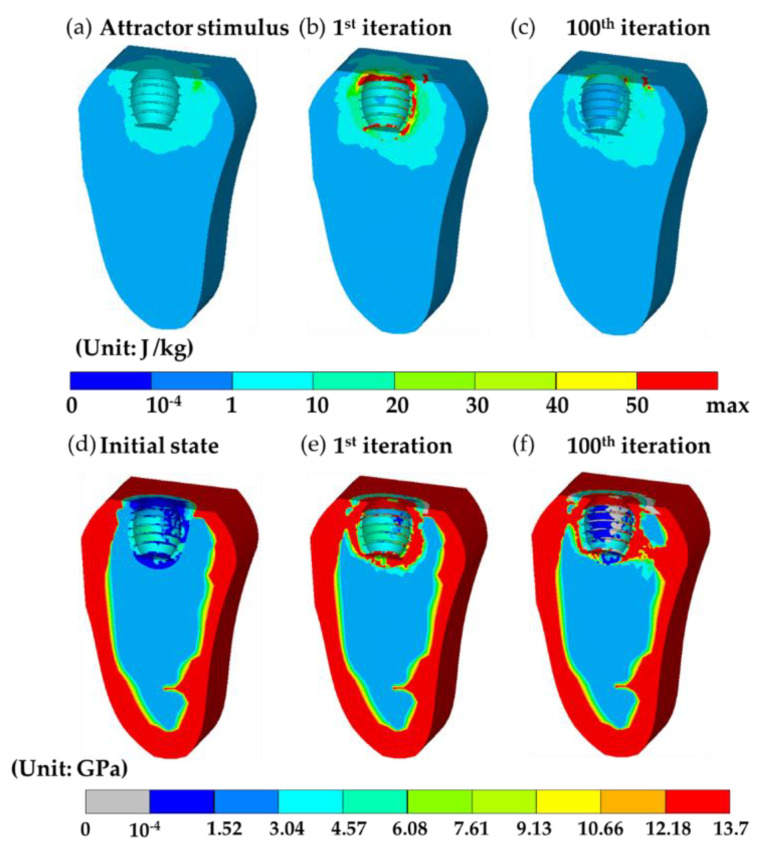
The results generated by the current model. (**a**) The target distribution of the attractor stimulus $K(\vec{r}$) based on the natural tooth. (**b**,**c**) are bone remodeling stimulus ($S_{BR}$ ) at the 1st and 100th iterations. (**d**–**f**) are the corresponding distribution of Young’s modulus during the bone remodeling process at the initial state, 1st, and 100th iteration.

**Figure 6 materials-13-02858-f006:**
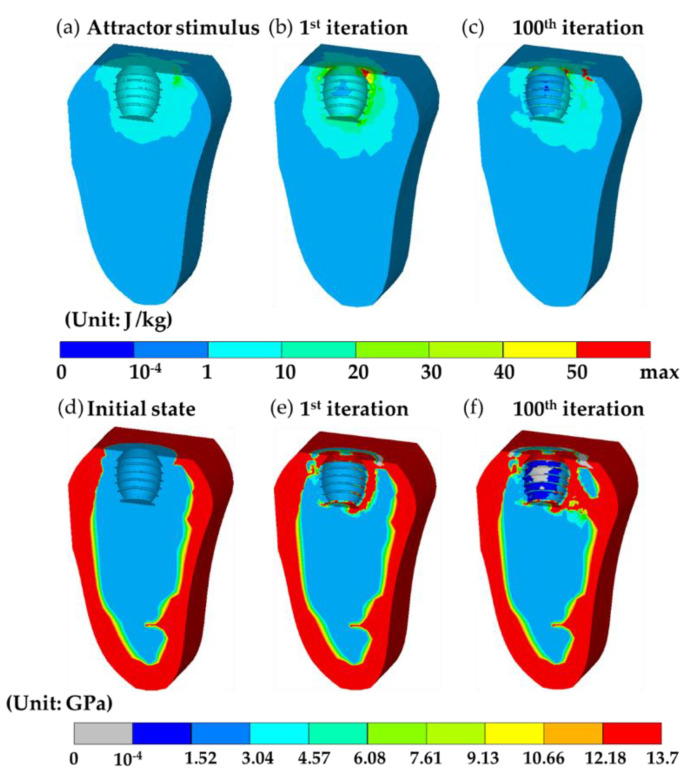
The results generated by the conventional model. (**a**) The target distribution of the attractor stimulus $K(\vec{r}$) based on the natural tooth. (**b**,**c**) are bone remodeling stimulus ($S_{BR}$
*)* at the 1st and 100th iterations. (**d**–**f**) are the corresponding distribution of Young’s modulus during the bone remodeling process at the initial state, 1st, and 100th iteration.

**Figure 7 materials-13-02858-f007:**
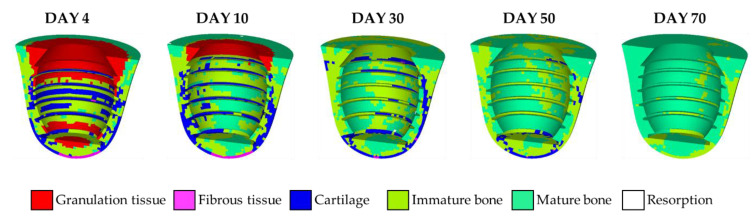
The tissue differentiation history predicted by the mechano-regulation algorithm in higher bone strength case.

**Figure 8 materials-13-02858-f008:**
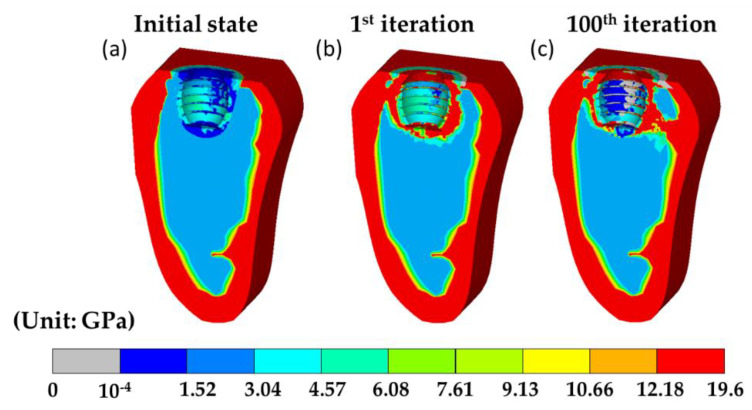
The results generated by the current model in higher bone strength case. (**a**–**c**) are the corresponding distribution of Young’s modulus during the bone remodeling process at the initial state, 1st, and 100th iteration.

**Table 1 materials-13-02858-t001:** The material properties used in the current work [29].

Materials	Young’s Modulus (GPa)	Poisson’s Ratio	Permeability (m^4^/Ns)
**Ti6Al4V**	113.8	0.34	N/A
**Tooth**	20	0.3	N/A
**Prosthesis**	80	0.3	N/A
**Bone graft**	2	0.3	N/A
**Cortical bone**	13.7	0.3	10^−17^
**Cancellous bone**	2	0.3	3.7 × 10^−13^
**Granulation tissue**	0.001	0.17	10^−14^
**Fibrous tissue**	0.002	0.17	10^−14^
**Cartilage**	0.01	0.17	5 × 10^−15^
**Immature bone**	1	0.3	10^−13^
**Mature bone**	6	0.3	3.7 × 10^−13^

N/A: not applicable.

**Table 2 materials-13-02858-t002:** The ranges of biophysical stimulus for different tissue phenotypes.

	$S_{TD}$		Tissue Phenotypes
3<	$S_{\mathrm{TD}}$		Fibrous tissue
1<	$S_{\mathrm{TD}}$	≤3	Cartilage
0.266<	$S_{\mathrm{TD}}$	≤1	Immature bone
0.010<	$S_{\mathrm{TD}}$	≤0.266	Mature bone
	$S_{\mathrm{TD}}$	≤0.010	Initial resorption

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
