# Peer review of "A Hybrid Model for Predicting Bone Healing around Dental Implants"

_materials, 2020, doi:10.3390/ma13122858_

Round 1
Reviewer 1 Report
The authors have reported a hybrid numerical bone healing algorithm to predict the morphology of bone around dental implants with the consideration of both short-term and long-term bone healing. The report is of potential value to the dental materials and implant dentistry fields. I recommend acceptance after enriching the results and discussion sections of the manuscript.
While the methodology section described the equations that form the algorithm, the results section seems to be lacking quantitative data. For example, Figure 3 shows that the tissue differentiation history gradually changed from granulation tissue and cartilage to mature bone. The percentage of coverage by various tissue types in each stage can be shown.
The bone resorption location and direction were discussed with references from the existing literature. How about the tissue differentiation history mentioned above?
Author Response
We thank the reviewers for the helpful comments. We have modified our manuscript accordingly, highlighting all changes in red. The reviewer raised some specific points we respond as follows:
- The authors have reported a hybrid numerical bone healing algorithm to predict the morphology of bone around dental implants with the consideration of both short-term and long-term bone healing. The report is of potential value to the dental materials and implant dentistry fields. I recommend acceptance after enriching the results and discussion sections of the manuscript.
A: Thanks for the suggestion. There are two additional paragraphs in “Results” and “Discussion”, where the issues of (1) percentage of various tissue phenotype in each day during the short-term healing process, and (2) the effect of bone with higher strength on bone healing were revealed and discussed.
- While the methodology section described the equations that form the algorithm, the results section seems to be lacking quantitative data. For example, Figure 3 shows that tissue differentiation history gradually changed from granulation tissue and cartilage to mature bone. The percentage of coverage by various tissue types in each stage can be shown.
A: Thanks for the reviewer’s suggestion. We have added Figure 3 and the corresponding description in “Results” (Sec. 3.1.), showing the percentage of tissue phenotypes in each day during the short-term healing process.
- The bone resorption location and direction were discussed with references from the existing literature. How about the tissue differentiation history mentioned above?
A: Thanks for the reviewer’s suggestion. We have added the texts in the 3rd paragraph in the discussion, addressing that the discussion about the sequence of tissue differentiation history by comparing it with both experimental and computational works in the literature.

Reviewer 2 Report
This article aims to propose a hybrid algorithm model to accurately predict bone healing after place a dental implant. The two steps model us based on short-term and long-term healing. The introduction gives a good overview of the current knowledge on the topic. The length is adequate and includes relevant literature. Material and methods are detailed described. The results could have a significant impact for clinical practice. However, I believe that there are some points that need to be clarified:
- What is the exact time point in which the authors considered short-time healing? In the same way, what is the exact point in which the authors considered long-time healing?
- 3D healing model is established on the second molar, could authors give more information about the sample studied?
- As it is known, there is a wide variety of people characteristics, and there are also several commercial materials currently used in dental reconstruction. What is the population scope of the hybrid model?
- Citations in the text need to be improved, using a single square bracket when more than one reference is cited.
- Please give details about the software suppliers.
- Line 60, “The algorithm was verified by CT images,…”. Please, explain here what does mean CT instead of line 98.
- Please, explain, in the figure legend, what does represent the yellow line in Figure 2b.
- What does mean Eq.? It is shown in line 151 as Eq. (2), line 156 as Eq. (3), line 176 as Eq. (5), line 179 as Eq. (6), line 180 as Eq. (6), line 185 as Eq. (7), line 204 as Eq. (3), line 234 as Eq. (8), and line 237 as Eq. (7).
Author Response
We thank the reviewers for the helpful comments. We have modified our manuscript accordingly, highlighting all changes in red. The reviewer raised some specific points we respond as follows:
- What is the exact time point in which the authors considered short-time healing? In the same way, what is the exact point in which the authors considered long-time healing?
A: Thanks for the reviewer’s question. We have added the texts in Line 165 and 199 to address this issue. The short-term healing time was set as 70 days, which is the average healing period for implantation surgeries [1–3]. The time step used in long-term healing is, in fact, a computational increment and is not associated with a real-life time scale. The end of the calculation depends on the convergence of Save as in the manuscript.
- 3D healing model is established on the second molar, could authors give more information about the sample studied?
A: The model of the current work was referred to as the mandibular second molar (back teeth in the upper jaw). We have added the texts in Line 101 accordingly. The bone geometry, density, and other material properties were adopted based on Chou et al. [4].
- As it is known, there is a wide variety of people characteristics, and there are also several commercial materials currently used in dental reconstruction. What is the population scope of the hybrid model?
A: Thanks for the reviewer’s suggestion. We have addressed this issue in the 4th paragraph in “Discussion.” Since the current hybrid model was implemented by the finite element method, which can perform virtual tests on a wide variety of people characteristics and dental materials by simply changing the geometry of the model, boundary conditions, and material properties of bones. Thus, the current model can potentially provide estimated information and may even provide optimized dental implants for dental clinics. We also examined a case of middle-aged male adults with higher strength material property of bone to demonstrate the applicability of the current model.
- Citations in the text need to be improved, using a single square bracket when more than one reference is cited.
A: Thanks for the reviewer’s suggestion. The citation style in the entire manuscript has been corrected.
- Please give details about the software suppliers.
A: Thanks for the reviewer’s suggestion. The software suppliers’ information have been added in Line 113. “The models of the two systems were built by a commercial finite element package ANSYS 18.0 (ANSYS, Inc., Canonsburg, Pennsylvania, USA)”
- Line 60, “The algorithm was verified by CT images,…”. Please, explain here what does mean CT instead of line 98.
A: Thanks for the reviewer’s suggestion. We have moved the term “computed tomographic (CT)” to Line 60.
- Please, explain, in the figure legend, what does represent the yellow line in Figure 2b.
A: Thanks for the reviewer’s suggestion. We have added a legend “cell origin” for the yellow line and the corresponding description in Line 152.
- What does mean Eq.? It is shown in line 151 as Eq. (2), line 156 as Eq. (3), line 176 as Eq. (5), line 179 as Eq. (6), line 180 as Eq. (6), line 185 as Eq. (7), line 204 as Eq. (3), line 234 as Eq. (8), and line 237 as Eq. (7).
A: We have revised all of the word “Eq.” into “Equation”.
Reference
- Monjo, M.; Lamolle, S.F.; Lyngstadaas, S.P.; Rønold, H.J.; Ellingsen, J.E.In vivo expression of osteogenic markers and bone mineral density at the surface of fluoride-modified titanium implants. Biomaterials 2008, 29, 3771–3780.
- Papalexiou, V.; Novaes Jr, A.B.; Grisi, M.F.M.; Souza, S.S.L.S.; Taba Jr, M.; Kajiwara, J.K.Influence of implant microstructure on the dynamics of bone healing around immediate implants placed into periodontally infected sites: A confocal laser scanning microscopic study. Clin. Oral Implants Res. 2004, 15, 44–53.
- JR, S.A.; Allegrini, M.R.F.; Yoshimoto, M.; JR, B.K.; Mai, R.; Fanghanel, J.; Gedrange, T.Soft tissue integration in the neck area of titanium implants--an animal trial. J. Physiol. Pharmacol. 2008, 59, 117–132.
- Chou, H.-Y.; Romanos, G.; Müftü, A.; Müftü, S.Peri-implant bone remodeling around an extraction socket: predictions of bone maintenance by finite element method. Int. J. Oral Maxillofac. Implants 2012, 27.

Reviewer 3 Report
Dear authors, thank you for this interesting paper. Your methodology is good and well explained, however, there are some issues I would like to point out so the quality of your research can be improved.
- Please check instructions for authors, citation style is not in the correct form, for example, [2]-[5] is wrong in your introduction.
Introduction:
2. 3rd paragraph "Most of the studies assumed..." Please provide references for this sentence.
3. Please place your objectives clearly at the end of the 5th (last) paragraph.
4. Please add null hypothesis
Materials and methods:
5. 1st paragraph: "several mathematical techniques". Please provide all methods/techniques used in the study
6. Point 2.1: When you talk about geometry of the bone, you say you extracted the characteristics of the bone from a CT image. But, what bone do you refer to? Bone geometry, density and other factors depend on the site of the mouth it belongs to, it is different for upper and lower jaw or for the bone surrounding front teeth and back teeth. Please clarify this issue.
7. Table 1. You give values for the different materials used. When you give values for the tooth, what tooth did you use as an example? are these values for incisors? molars? Please clarify. The same with bone (depends on the part of the mouth as I commented in point 6) or for bone graft (which type of graft). Please clarify these issues.
8. Point 2.2 line 157. "70 days". Please state that you analyzed the model at 4, 10, 30, 50 and 70 days.
9. Fig. 3. No need to add letters (a, b, c...)
Discussion
10. Discussion is poor, please discuss your methods and results further and add comparisons with other authors.
11. Please discuss different types of bone, teeth, grafts etc and how this could affect your results
Conclusions
11. Conclusions are too long, they need to be more concise and they need to answer your objectives point by point
Author Response
We thank the reviewers for the helpful comments. We have modified our manuscript accordingly, highlighting all changes in red. The reviewer raised some specific points we respond as follows:
- Please check instructions for authors, citation style is not in the correct form, for example, [2]-[5] is wrong in your introduction.
A: Thanks for the reviewer’s suggestion. The citation style in the entire manuscript has been corrected.
Introduction:
- 3rd paragraph "Most of the studies assumed..." Please provide references for this sentence.
A: We have added the following four references to support the statement.
- Chou, H.-Y.; Romanos, G.; Müftü, A.; Müftü, S.Peri-implant bone remodeling around an extraction socket: predictions of bone maintenance by finite element method. Int. J. Oral Maxillofac. Implants 2012, 27.
- Lian, Z.; Guan, H.; Ivanovski, S.; Loo, Y.-C.; Johnson, N.W.; Zhang, H.Effect of bone to implant contact percentage on bone remodeling surrounding a dental implant. Int. J. Oral Maxillofac. Surg. 2010, 39, 690–698.
- Lin, D.; Li, Q.; Li, W.; Swain, M.Bone remodeling induced by dental implants of functionally graded materials. J. Biomed. Mater. Res. Part B Appl. Biomater. An Off. J. Soc. Biomater. Japanese Soc. Biomater. Aust. Soc. Biomater. Korean Soc. Biomater. 2010, 92, 430–438.
- Li, J.; Li, H.; Shi, L.; Fok, A.S.L.; Ucer, C.; Devlin, H.; Horner, K.; Silikas, N.A mathematical model for simulating the bone remodeling process under mechanical stimulus. Dent. Mater. 2007, 23, 1073–1078.
- Please place your objectives clearly at the end of the 5th (last) paragraph.
A: Thanks for the reviewer’s suggestion. We have revised the last paragraph in the introduction to specifying the objective of the current work.
- Please add a null hypothesis
A: Thanks for the reviewer’s suggestion. We have added texts in line 75 and addressed that the conventional model can be served as a null hypothesis which assumed uniform material properties around the implants, i.e. the effect of short-term tissue differentiation was not considered.
Materials and methods:
- 1st paragraph: "several mathematical techniques". Please provide all methods/techniques used in the study
A: Thanks for the reviewer’s suggestion. We are sorry for such confusion. These mathematical techniques (i.e. rule of mixture and smoothing procedure) have been explained in detail in Sec. 2.2. We have revised the texts as follows to eliminate the confusion. “Next, the rule of mixture and smoothing procedure [36] was applied to determine the updated material properties and the detail will be discussed in Sec. 2.2.”
- Point 2.1: When you talk about geometry of the bone, you say you extracted the characteristics of the bone from a CT image. But, what bone do you refer to? Bone geometry, density and other factors depend on the site of the mouth it belongs to, it is different for upper and lower jaw or for the bone surrounding front teeth and back teeth. Please clarify this issue.
A: The model of the current work was referred to as the mandibular second molar (a back teeth in the upper jaw). We have added the texts in Line 101 accordingly. The bone geometry, density, and other material properties were adopted based on Chou et al. [1].
- Table 1. You give values for the different materials used. When you give values for the tooth, what tooth did you use as an example? are these values for incisors? molars? Please clarify. The same with bone (depends on the part of the mouth as I commented in point 6) or for bone graft (which type of graft). Please clarify these issues.
A: Thanks for the reviewer’s suggestion. We have added the texts in Line 123, addressing that loading, setting, and properties of all materials, including bones, prosthesis, tooth, implant, and bone graft, used in the current work were based on Chou et al. [1] for comparison.
- Point 2.2 line 157. "70 days". Please state that you analyzed the model at 4, 10, 30, 50 and 70 days.
A: Thanks for the reviewer’s suggestion. We are sorry for such confusion. The results of tissue differentiation are a continuous process, where the entire process was calculated iteration by iteration (i.e. day by day). In order to clarify the confusion, we have added Figure 3 and the corresponding description in the Results (Sec. 3.1.), showing the percentage of tissue phenotypes in each day during the short-term healing process.
- Fig. 3. No need to add letters (a, b, c...)
A: Thanks for the reviewer’s suggestion. We have removed all the letters in the Figure.
Discussion
- Discussion is poor, please discuss your methods and results further and add comparisons with other authors.
A: Thanks for the reviewer’s suggestion. We have added the texts in the 3rd paragraph in the discussion and made comparisons with other authors.
- Please discuss different types of bone, teeth, grafts etc. and how this could affect your results
A: Thanks for the reviewer’s suggestion. We adopted the case of middle-aged male adults with higher strength material property of bone, referred to as to higher bone strength case to reveal the effect of different types of bone. The results were compared with the original standard case and discussed in the section of Discussion. It is worth to mention that the effect of teeth and grafts was discussed in other literature, such as Chou et al. [1]. Therefore, these topics were not addressed in detail in the current work.
Conclusions
- Conclusions are too long, they need to be more concise and they need to answer your objectives point by point.
A: We have shortened our conclusion from 204 words into 106 words, in order to be more concise.
Reference
1. Chou, H.-Y.; Romanos, G.; Müftü, A.; Müftü, S.Peri-implant bone remodeling around an extraction socket: predictions of bone maintenance by finite element method. Int. J. Oral Maxillofac. Implants 2012, 27.

Round 2
Reviewer 1 Report
The authors have addressed my concerns.
Author Response
We thank the reviewers for the previous useful comments.
Reviewer 2 Report
The authors have provided satisfactory the comments addressed.
Author Response
We thank the reviewers for the previous helpful comments.
Reviewer 3 Report
Dear authors, thank you for providing a revised version of your manuscript. Most of the requested changes have been made. However, there are still a couple of issues I would like you to revise:
- Null hypotesis is still not clear. Please explain better and clarify
- Please add the acceptance/rejection statement of your null hypothesis to the discussion section.
Author Response
We thank the reviewers for the helpful comments. We have modified our manuscript accordingly, highlighting all changes in red. The reviewer raised a specific point and we respond as follows:
- Null hypothesis is still not clear. Please explain better and clarify
Please add the acceptance/rejection statement of your null hypothesis to the discussion section.
A: Thanks for the reviewer’s suggestions. We have explained the statement of our null hypothesis more clearly in Line 61-67, and reject the null hypothesis in Line 354 based on the features discovered by our hybrid model.